# The Effect of Genomic DNA Contamination on the Detection of Circulating Long Non-Coding RNAs: The Paradigm of *MALAT1*

**DOI:** 10.3390/diagnostics11071160

**Published:** 2021-06-25

**Authors:** Athina N. Markou, Stavroula Smilkou, Emilia Tsaroucha, Evi Lianidou

**Affiliations:** 1Analysis of Circulating Tumor Cells Lab, Lab of Analytical Chemistry, Department of Chemistry, National and Kapodistrian University of Athens, 15771 Athens, Greece; stavroulasmilkou2395@gmail.com (S.S.); evilianidou@gmail.com (E.L.); 28th Department of Pulmonary Diseases, ‘Sotiria’ General Hospital for Chest Diseases, 11527 Athens, Greece; emilygeola@yahoo.gr

**Keywords:** *MALAT1*, non-long coding RNA, DNAse treatment, cfRNA, tumor biomarkers, false positive results, gDNA

## Abstract

The presence of contaminating gDNA in RNA preparations is a frequent cause of false positives in RT-PCR-based analysis. However, in some cases, this cannot be avoided, especially when there are no exons–intron junctions in the lncRNA sequences. Due to the lack of exons in few of long noncoding RNAs (lncRNAs) and the lack of DNAse treatment step in most studies reported so far, serious questions are raised about the specificity of lncRNA detection and the potential of reporting false-positive results. We hypothesized that minute amounts of gDNA usually co-extracted with RNA could give false-positive signals since primers would specifically bind to gDNA due to the lack of junction. In the current study, we evaluated the effect of gDNA and other forms of DNA like extrachromosomal circular DNAs (eccDNAs) contamination and the importance of including a DNAse treatment step on lncRNAsexpression.As a model, we have chosen as one of the most widely studied lncRNAs in cancer namely *MALAT1*, which lacks exons. When we tested this hypothesis in plasma and primary tissue samples from NSCLC patients, our findings clearly indicated that results on *MALAT1* expression are highly affected by the presence of DNA contamination and that the DNAse treatment step is absolutely necessary to avoid false positive results.

## 1. Introduction

Non-coding RNAs (ncRNAs) are RNA molecules that are not translated into a protein but their functions are undoubtedly crucial in several mechanisms. They are divided into short ncRNAs and long ncRNAs(lncRNAs) based on their nucleotide length. The lncRNAs are non-protein-coding transcripts with a length over 200 nucleotides (nt), and consist of the broadest class of ncRNAs [1]. In a meta-analysis, Iyer et al. showed that from a consensus of around 91,000 human genes, over 68% of genes were classified as lncRNAs, of which 79% were previously unannotated [2]. During the last decades, lncRNAs were found to be involved in many biological processes [3] and have thus gained considerable interest as principal regulators of gene expression in several different ways [4,5]. In addition, numerous studies have shown that lncRNAs are abnormally expressed in many cancers, such as breast, lung and prostate [6,7].

Various total RNA and cfRNA isolation methods have been described for the extraction of non-long coding RNAs that are further detected by molecular techniques like RT-qPCR, RNA sequencing, and FISH analysis. Co-isolation of genomic DNA (gDNA), and other forms of DNA, like extrachromosomal circular DNAs (eccDNAs) during the extraction of total RNA from plasma and tissues, is inevitable, unless a DNAse treatment step is included prior to RT-PCR. The presence of contaminating DNAs in RNA preparations can cause false positives in RT-PCR in case those primers are fully overlapping with gDNA sequences. To avoid gDNA co-amplification, specific precautions must be taken in the primers’ design. However, in some cases, this cannot be avoided, especially when the target mRNA presents pseudogenes at the DNA level or when there are no exons–intron junctions.

Many of the well-studied lncRNAs, that are considered as regulatory molecules with various significant functions in cancer, have no junctions in their sequences such as *MALAT1*, *NKILA*, *NEAT1* and *NORAD* [8,9,10,11] (Table 1).

Thus, RNA analysis in clinical samples could lead to false-positive results, due to gDNA contamination, and the lack of exons in few of the lncRNAs and the overlapping of all primers designed with gDNA. We noticed that in the vast majority of lncRNAs studies, the expression levels of lncRNAs were evaluated by RT-qPCR without taking into account the possibility of false-positive results due to gDNA contamination. For this reason, we aimed to examine this by analyzing clinical samples for Metastasis-Associated Lung Adenocarcinoma Transcript 1 (*MALAT1)* with and without DNAse treatment.

*MALAT1* is one of the most widely studied lncRNAs in cancer. *MALAT1,* located on chromosome 11q13 and especially on nuclear speckles [12], was initially identified as a prognostic marker in non-small-cell lung cancer (NSCLC) [13]. Nowadays, its function as an oncogene has been evaluated in several types of cancer like colorectal [14], ovarian [15,16] and gastric cancer [17]. *MALAT**1* has also been proposed as a reliable biomarker, not only for diagnosis and prognosis, but also in targeted therapy for leukemia [18,19]. Interestingly, the most abundant transcript variant of *MALAT1* is the long variant (NR_002819), which lacks exons. As expected, all studies published so far on *MALAT1* in cancer use primers that co-hybridize to genomic DNA (Figure 1). However, in the majority of these studies, *MALAT1* expression is evaluated by RT-qPCR without any prior DNAse treatment for the removal of contaminating gDNA (Table 2), thus giving rise to significant concerns on specificity, and the presence of false positives, which is highly crucial for the clinical significance of the results presented.

In the current study, we evaluated, for the first time, the effect of gDNA contamination and the importance of including a DNAse treatment step on the expression levels of one of the most widely studied lncRNAs, *MALAT1*. We tested this in plasma and primary tissue samples from NSCLC patients and healthy donors. Our findings clearly indicated that most results reported so far on *MALAT1* expression are highly affected by gDNA contamination, and this could be also extrapolated to all lncRNAs without exons.

## 2. Materials and Methods

### 2.1. Patients and Samples

We analyzed a total of 48 clinical samples: (i) 15 peripheral blood samples from patients with early NSCLC, (ii) 15 peripheral blood samples from healthy donors, and (iii) 9 primary tissues of surgically resected NSCLC and their adjacent noncancerous tissue specimens. All patients gave their informed consent, and the Ethical and Scientific Committees of the participating institutions approved the study (28872/10-12-19). At the time of surgery, all tissue samples were immediately flashfrozen in liquid nitrogen and stored at −70 °C until use. We analyzed all samples histologically to assess the amount of tumor component (at least 70% tumor cells) and the quality of material (i.e., absence of necrosis).

### 2.2. Plasma Preparation

Peripheral blood samples (30 mL) isolated in K_3_-EDTA tubes were centrifuged at 530× *g* for 10 min at room temperature, without brakes, within 6 h after collection. Plasma was transferred to fresh tubes and centrifuged at 2000× *g* for 10 min. Finally, plasma was divided into 2 mL aliquots in fresh tubes, and stored at −80 °C. All samples were collected in the morning before surgery from early NSCLC patients.

### 2.3. RNA Extraction

Circulating cell-free RNA (ccfRNA) was extracted from 600 μL of plasma using miRNeasy Serum/Plasma Advanced Kit (Qiagen, Hilden, Germany) according to the manufacturer’s instructions, with an elution volume of 25 μL in RNase-free water. In tissue samples, total cellular RNA isolation was performed using the QiagenRNeasy Mini Reagent kit (Qiagen, Hilden, Germany)according to the manufacturer’s instructions [36]. All preparation and handling of RNA took place in a laminar flow hood, under RNase-free conditions, and the isolated RNA was stored at −70 °C until use. RNA concentration was determined with a NanoDrop ND-100 spectrophotometer (NanoDrop Technologies). To accurately assess sample quality, 260/280 and 260/230 ratios were analyzed in combination with overall spectral quality and the yield of 260/280 ratio was acceptable at ~2.0 for RNAs. RNA of each sample was spilt into two aliquots of 10 μL each.

### 2.4. DNAse Treatment

Initially, in each reaction tube were added: ≤200 ng/μL input RNA, 1 μL of TURBO DNAse Buffer (Ambion Life Technologies, Austin, TX, USA) and 0.4U DNAse I enzyme (Ambion Life Technologies, Austin, TX, USA), and the sample was incubated at 37 °C for 20min. One microliter of DNAse inactivation reagent was then added for 5min followed by centrifugation at 10,000× *g* for 1.5 min. The supernatant which contains the RNA was then carefully transferred into a fresh tube. The whole procedure was done under DNAse-free conditions to avoid DNA contamination (dedicated specific lab areas, labware, laminar-flow hood). In order to optimize the DNase treatment step, RT-PCR for *MALAT1* and *B2M* in different concentrations of gDNA before and after treatment was performed. Due to the lack of exons in MALAT1, primers would specifically bind to gDNA but B2M primers were designed in the junction area; there was no influence of the presence of gDNA and DNase treatment. Complete degradation of DNA was considered when there was no detection of *MALAT1* after DNase treatment in gDNA samples (Figure 2).

### 2.5. cDNA Synthesis

The high-capacity RNA-to-cDNA kit (Applied Biosystems, Foster City, CA, USA) was used for reverse transcription in 20 μL of total volume reaction. A negative control was included in each experiment to ensure that there was no contamination by genomic DNA (gDNA). All cDNA samples were stored at −20 °C until further molecular analysis.

### 2.6. RT-qPCR Assay

We first designed insilicohighly specific primers for *MALAT1* based on its RNA sequence (NR_002819.4) using Primer Premier 5.0 software (Premier Biosoft, San Francisco, CA, USA). The designed primers for *MALAT1* are the following: forward: 5′-CCCCACAAGCAACTTCTCTG-3′ and, reverse: 5′-TCCAAGCTACTGGCTGCATC-3′. The experimental conditions of RT-qPCR for *MALAT1* expression were optimized (annealing temperature, time, primer MgCl_2_, dNTPs, and BSA concentrations). Each reaction was performed in the LightCycler^®^ 2.0 System (IVD instrument, Roche Diagnostics, Mannheim, Germany) in a total volume of 10 μL, following the MIQE guidelines [37]. One microliter of cDNA was added to a 9 μL reaction mixture. The amplification reaction for *MALAT1* contained 2 μL of the PCR Synthesis Buffer (5Χ), 1 μL of MgCl_2_ (25 mM), 0.2 μL dNTPs (10 mM), 0.5 μL BSA (10 μg/μL), 0.1 μL Hot Start DNA polymerase (HotStart, 5 U/μL, Promega, Dane County, WI, USA), 0.3 μL of forward and reverse primer (10 μM), 1 μL of 1X LC Green^®^ (Idaho Technology, Salt Lake City, UT, USA). RT-qPCR protocol begins with one cycle at 95 °C for 2 min followed by 45 cycles of 95 °C for 10 s, 60 °C for 10 s, and 72 °C for 10 s. Immediately after amplification, a rapid cooling cycle to 40 °C for 30 s was introduced in order to prepare the melting curve acquisition step. Real-time fluorescence acquisition was set at the elongation step (72 °C). The following melting curve analysis included the steps of 55 °C for 20 s, 95 °C for 0 s with a ramp rate 0.19 °C/s (acquisition mode: continuous), and 40 °C for 10 s. Additionally, we used our previously developed and analytically validated RT-qPCR assays for beta-2-microglobulin (*B2M*), used as a reference gene [38]. In each RT-qPCR run, we used the same cDNA from MCF-7 cells as a positive control in order to evaluate the accuracy and reproducibility of the results.

### 2.7. Normalization of Data

Expression values of *MALAT1* were normalized to *B2M*. ΔCq values were calculated by using Cq values for *MALAT1* and the corresponding *B2M* for each sample. We calculated ΔΔCq values using ΔCq values for cancerous samples and the mean value of ΔCq for normal samples (ΔΔCq = ΔCqcancer − Δcqnormal). Relative quantification (RQ) was based on the ΔΔCq method as described [39]. For paired tissue samples, ΔCq values were calculated as the differences between ΔCq values for each cancerous sample and its corresponding adjacent normal tissue. *MALAT1* expression data are presented as fold change relative to the reference gene based on the formula of RQ = 2^−ΔΔCq^.

### 2.8. Statistical Analysis

We performed statistical evaluation of data using SPSS (SPSS Statistics version 26.0). A level of *p* < 0.05 was considered statistically significant. Statistical analysis was performed in all cases by using paired sample *t*-test.

## 3. Results

The experimental flowchart of the study is outlined in Figure 3.

### 3.1. Optimization of DNase Treatment Conditions

#### 3.1.1. Enzyme Incubation

Equal amounts of total RNA (200 ng) were either treated with TURBO DNA-free™ at different incubation times (5, 10, and, 20 min) at 37 °C, or were left untreated. Treated and untreated RNA samples were reverse-transcribed and cDNA were further analyzed by RT-PCR. RT-PCR was first performed for *B2M* as reference gene in order to ensure that treatment of RNA with TURBO DNA-free™ maintains target sensitivity in real-time RT-PCR. RNA quality as estimated through *B2M* expression is not affected when DNAse treatment was performed for 5 or 10 min in 37 °C (Figure 4a). All experiments were run in triplicate.

#### 3.1.2. Concentration of gDNA

We evaluated the effectiveness of the DNAse treatment step by analyzing gDNA samples at concentrations of 20 ng/μL and 5 ng/μL, using 5 and 10 min as incubation time. gDNA was added in the same RNA samples that were split into two aliquots of 10 μL each. One aliquot was treated with TURBO DNA-free™ (for 5 and 10 min of enzyme incubation) and the other aliquots were left untreated. We found out that DNAse incubation at 37 °C for 5 min is not enough for the complete elimination of gDNA. However, when we increased the DNAse incubation time for 10 min, there was no signal for gDNA, while at the same time the effect of DNAse treatment in the quality of RNA was limited (Figure 4b).

#### 3.1.3. Repeatability of the Procedure

We evaluated the repeatability of the whole procedure (within run imprecision) by analyzing the same RNA sample in 3 parallel determinations after DNAse treatment and without DNAse treatment. Intra-assay CVs were satisfactory in all cases.

### 3.2. False-Positive Results on MALAT1 Expression in Clinical Samples

#### 3.2.1. NSCLC Primary Tissues

We compared the expression of *MALAT1* in nine pairs of NSCLC tissues and their adjacent noncancerous tissues using RT-qPCR in samples with and without DNAse treatment. *MALAT1* expression was normalized with respect to *B2M* gene expression based on the relative quantification approach [23]. We observed that in untreated samples, *MALAT1* was found to be overexpressed in 6/9 (66.7%) NSCLC tissues, while only 3/6 (50%) remained positive for *MALAT1* after DNAse treatment, showing that, in total, 3/9 (30%) tested samples were false positive before the DNAse treatment step. *MALAT1* was found to be downregulated in 3/9 (33.3%) tested paired samples before treatment but only 1/9 (11.1%) remained downregulated after DNAse treatment. In 4/9 (44.4%) paired samples, there was no differentiation in the expression of *MALAT1* (Appendix A). More specifically, in 3/4 (75%) of the samples, we detected *MALAT1* overexpression both with and without DNAse treatment, while in 1/4 (25%) of the samples we detected lower expression (Table 3).

#### 3.2.2. ccfRNA in Plasma

*MALAT1* and *B2M* expression were evaluated in 30 RNA samples directly isolated from plasma of early NSCLC patients (*n* = 15) and healthy donors (HD) (*n* = 15). Without DNAse treatment, *MALAT1* was detected in all samples and overexpression was observed in 3/15 (20%) of NSCLC patients (Appendix A). Interestingly, after DNAse treatment, *MALAT1* was detected only in 4 of the tested samples and none of them was overexpressed in *MALAT1* (Table 3), proving once again the effect of gDNA and the detection of false-positive results.

## 4. Discussion

lncRNAs have been evaluated as novel tumor biomarkers, not only in diagnosis and prognosis, but also in targeted therapy for different types of cancer [40,41]. RT-qPCR is extensively used for the quantification of lncRNAs transcripts. Contamination of gDNA and other forms of DNAs like extrachromosomal circular DNAs (eccDNAs)—which are the major form of extrachromosomal DNAs—is an inherent problem during RNA purification due to the similar physicochemical properties of RNA and DNA [42,43].

We report in this study that false-positive results could arise due to gDNA contamination and overestimate the abundance of lncRNAs transcript levels. RT-PCR assays can be designed to be gDNA insensitive only if primers can be designed. Such as those designed to target exons flanking a long intron or with primers that cross exon–exon junctions. It is expected that all RT-PCR assays for single-exon genes, like in the case of several lncRNAs, will readily amplify contaminating gDNA.

NEAT1overexpression was associated with poor prognosis in several types of cancer, like breast [44] and digestive system tumors [9], and it was suggested that it could be used as a promising biomarker for diagnosis [45]. NORAD is another lncRNA, which is reported to be overexpressed in many cancers and several studies have explored its involvement in numerous processes associated with carcinogenesis [10]. Moreover, *NKILA* underexpression was shown to be an effective prognostic and diagnostic biomarker in human cancer [8,46]. The expression of *MALAT1* has been evaluated in numerous studies and its increased expression has been correlated with poor overall survival in patients with solid malignancies [47,48]. A common characteristic of all these lncRNAs is the lack of exons in their sequences. Intriguingly almost all studies that have evaluated the expression of these lncRNAs do not include any DNAse treatment step, and thus there is a high probability of reporting falsepositive results.

In this study, we evaluated for the first time the effect of gDNA on the expression levels of *MALAT1,* a well-studied lncRNA that has a single exon, using RT-qPCR. We initially optimized the protocol to achieve specific DNAse treatment with the lowest effect on RNA and further compared *MALAT1* expression in clinical samples before and after DNAse treatment. Our findings clearly indicate that the expression levels of *MALAT1* were significantly affected by the presence of gDNA. In paired NSCLC tissue samples, we observed a significant difference in the expression of *MALAT1* before and after DNAse treatment in the majority of samples (56.6%). It is highly important that *MALAT1* expression was not detected in 73.4% of plasma samples after DNAse treatment. Τhis observation, combined with the lack of DNAse treatment, may explain the ambiguous results of various studies that characterized *MALAT1* either as oncogene or as tumor suppressor and consequently report that its expression is upregulated or downregulated, respectively [21,30,49,50]. One of the few studies that performed DNAse treatment before quantification of *MALAT1* expression demonstrated that *MALAT1* is a metastasis-suppressing lncRNA rather than a metastasis promoter lncRNA in breast cancer [24].

In conclusion, we report for the first time that contamination of gDNA can seriously affect lncRNAs expression results and cause false positives. Our findings need to be further evaluated and validated in a large and well-defined patient cohort. Taking into account that lncRNAs have gained widespread attention in recent years as potentially new and crucial candidates for tumor biomarkers, we conclude that DNAse treatment is a mandatory step in cases where there are no exons in lncRNAs sequence in order to ensure specificity. It is only under these conditions that the clinical significance of lncRNAs will be reliably revealed.

## Figures and Tables

**Figure 1 diagnostics-11-01160-f001:**
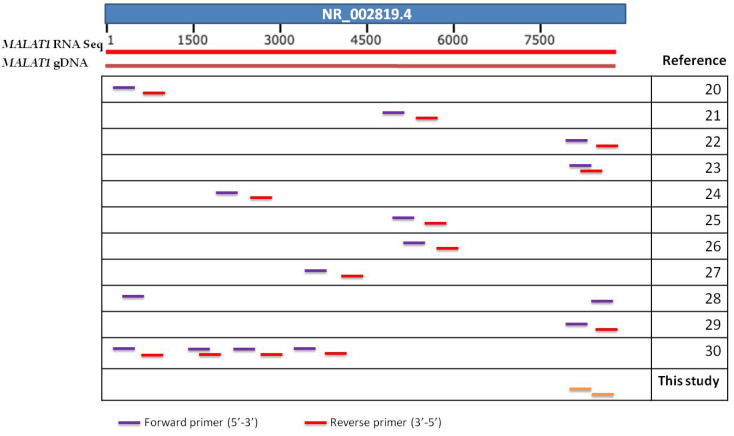
Graphic summary of lncRNA *MALAT1* reference sequence, *MALAT1* gDNA and position of different pairs of primers designed used in various studies. The last pair (green) was the one designed in the present study.

**Figure 2 diagnostics-11-01160-f002:**
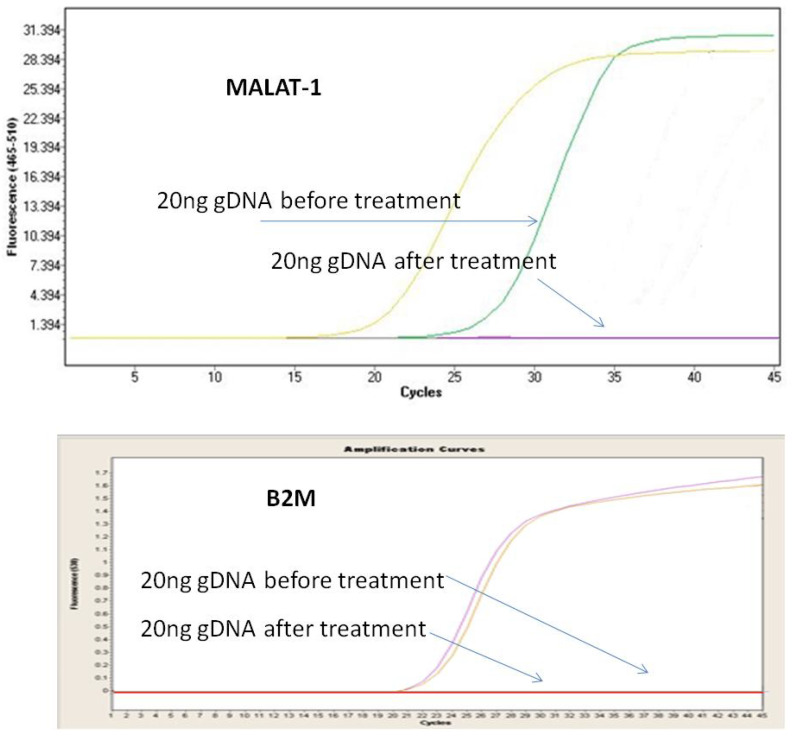
B2M and MALAT1 expression before and after DNase treatment.

**Figure 3 diagnostics-11-01160-f003:**
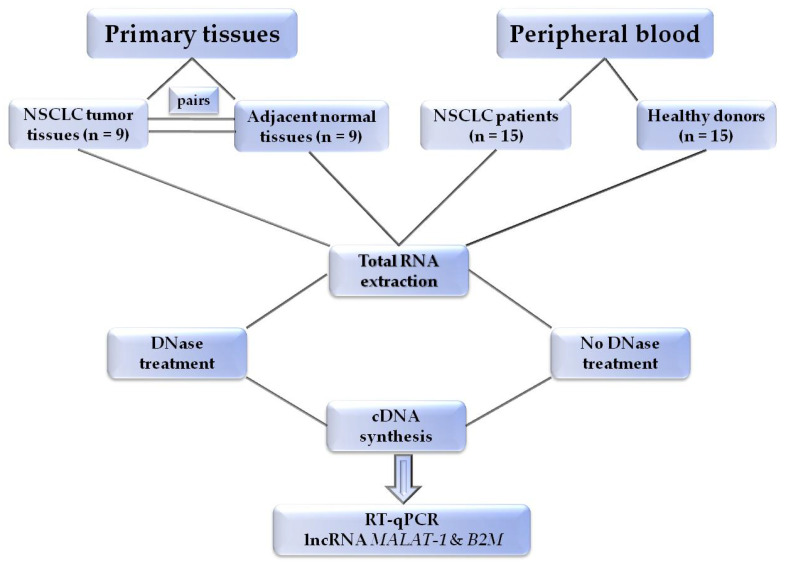
Outline of the experimental procedure.

**Figure 4 diagnostics-11-01160-f004:**
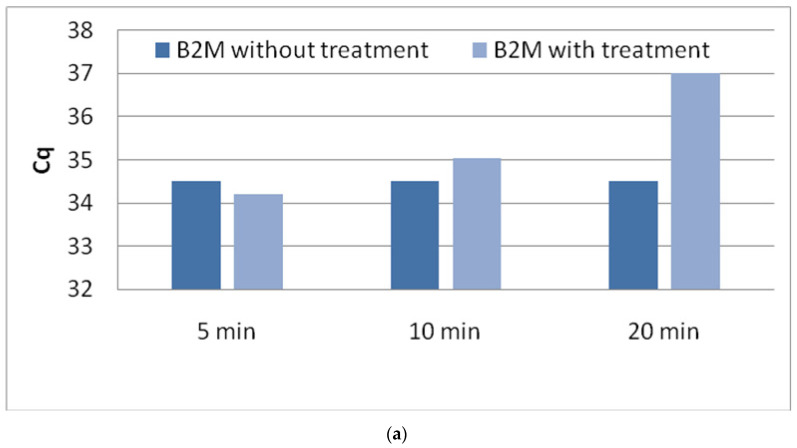
(**a**): Effect of enzyme incubation time. (**b**):Effect of gDNA concentration.

**Table 1 diagnostics-11-01160-t001:** Long non-coding RNAs which lack exons.

Long Non-Coding RNA	Cancer Types	Ref. Sequence
*MALAT1*	Lung cancer, Esophageal carcinoma, Acute myeloid leukemia, Ovarian cancer, thyroid, nerve—tibial, skin, uterus, prostate	NR_002819.4
*NEAT1*	Breast, lung, Prostate cancer, Head and neck squamous cell carcinoma, colon cancer, thyroid cancer	NR_028272.1
*DLEU*	Acute myeloid leukemia, spleen, lung cancer, esophageal carcinoma, pancreatic, laryngeal, renal, cervical cancer	NR_002612.1
*ANRASSF1*	Breast, prostate, astric cancer	NR_109831.1
*NKILA*	Pancreatic adenocarcinoma, prostate, breast cancer, uterine carcinosarcoma, lung cancer	NR_131157.1
*NORAD*	Lymph node metastasis, pancreatic, bladder, gastric cancer; esophageal squamous cell carcinoma, epatocellularcarcinoma, colorectal cancer	NR_027451.1
*KCNQ1OT1*	Esophageal carcinoma, Acute myeloid leukemia, Ovarian, Stomach adenocarcinoma	NR_002728.3
*CCAT2*	Colon cancer, breast cancer, hepatocellular carcinoma	NR_109834.1
*LincRNA-p21*	Prostate, gastric, colorectal cancer, head and neck squamous cell carcinoma, lung cancer	CD515754.1

**Table 2 diagnostics-11-01160-t002:** MALAT1 expression in cancer.

Cancer Type	Sample Origin	Number of Samples	Reference Gene	DNase Treatment	Expression of MALAT-1	Significance	Reference
NSCLC	Tissues (tumor and adjacent) Cell Lines	40	*GAPDH*	**Yes**	Over-expressed	Therapeutic target	[20]
	Plasma Tissues	105 65	*GAPDH*	No	Under-expressed	Diagnostic	[21]
	Tissues (tumor and adjacent)	86	*GAPDH*	No	Over-expressed	Tumor progression and development	[22]
	Serum (exosomes)	77	*GAPDH*	No	Over-expressed	Diagnostic, prognostic, therapeutic target	[23]
	Plasma	142	*18S rRNA*	No	Over-expressed	Diagnosis of EGFR-mutant patients	[24]
	Cell Lines Tissues (tumor and adjacent)	42	*RNU6B*	No	Over-expressed	Therapeutic target	[25]
	Cell Lines Tissues (tumor and adjacent)	30	*GAPDH or U6*	No	Over-expressed	Diagnostic	[26]
	Cell Lines Tissues (tumor and adjacent)	36	*GAPDH*	No	Over-expressed	Therapeutic target	[27]
Prostate cancer	Plasma Tissues (tumor and adjacent)	169 14	*β*-*actin*	**Yes**	Over-expressed	Diagnostic	[28]
	Cell Lines Tissues (tumor and adjacent) Mice	52	*β*-*actin*	No	Over-expressed	Therapeutic target	[29]
Breast cancer	Cell Lines Mice	-	*GAPDH*	**Yes**	Under-expressed	Prognostic, Therapeutic target	[30]
	Cell Lines Mice	-	*YWHAZ*	No	Under-expressed	Therapeutic target	[31]
	Cell Lines Clinical samples	-	GAPDH	No	Over-expressed	Prognostic	[32]
Gastric cancer	Tissues Plasma	64	*β*-*actin*	No	Over-expressed	Prognostic, diagnostic	[33]
	Cell Lines Tissues (tumor and adjacent)	57	*GAPDH*	No	Over-expressed	Therapeutic target	[34]
	Cell Lines Tissues Mice	153	*GAPDH*	No	Over-expressed	Therapeutic	[35]

The bold “Yes” means that in this study DNase treatment was performed.

**Table 3 diagnostics-11-01160-t003:** *MALAT1* expression in plasma (*n* = 15) and tissue samples (*n* = 9 pairs) of NSCLC patients before and after DNase treatment.

Plasma			
After DNAse treatment		Before DNAse treatment	
		Overexpression	Underexpression
	Overexpression	0	0
	Underexpression	3 (False positive)	12
		Paired *t*-test: 0.082	
		False positives: 3/15 (20%)	
Primary Tissues			
After DNAse treatment		Before DNAse treatment	
		Overexpression	Underexpression
	Overexpression	3	2
	Underexpression	3 (False positive)	1
		Paired *t*-test: 0.681	
		False positives: 3/9 (30%)	

## Data Availability

The data presented in this study are available on request from the corresponding author.

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
