# Peer review of "The Effect of Genomic DNA Contamination on the Detection of Circulating Long Non-Coding RNAs: The Paradigm of MALAT1"

_diagnostics, 2021, doi:10.3390/diagnostics11071160_

Round 1
Reviewer 1 Report
Comments to authors:
The manuscript that was written by Dr. Markou et al. is interesting to show that expression of lncRNA should be analyzed after DNase treatment to avoid false positive results. Authors performed RT-qPCR analysis with total RNA-containing samples after DNase treatment and reported that overestimation of expression of MALAT1 could be caused by the contained gDNAs in the RNA samples used for the analysis. Thus, authors are warning that samples for RT-qPCR should be treated with DNase. The finding should be applicable on the analysis of non-intronic gene expression. However, before publishing the article, authors should confirm and describe the experimental procedures are appropriate scientifically.
General comments
In this study, clinical samples were used for the lncRNA analysis. If the experiments have been examined and approved by some committee for human clinical experiments, it should be noted in the text. Moreover, it is not described how were the conditions and quality of the samples. The quality of RNAs should be evaluated before the analysis. Presumably, absorbance of 260nm/280nm might have been analyzed, how about the degradation of RNAs? I am afraid if all the samples were applicable to the analysis or not. Because this research article is focused on the analysis on nucleotides, the analytical data to show the qualities of samples are essential.
Specific comments
Materials and methods
2.4. DNase treatment:
In this experimental condition, it is described that 0.4 U of DNase I at 37 ℃ for 20 min was enough to degrade DNAs completely. How was that evaluated? Did authors perform reactions with internal control, such as purified plasmid DNAs or some specific DNAs-containing DNase reaction? The condition-determining experimental results should be essential for the discussion.
Results
3.1.2 Concentration of gDNA:
To show the reason why authors evaluated the incubation time for 10 min was enough, but 5 min was not, experimental data should be shown.
In Figure 3a, B2M without treatment and B2M with treatment. On the other hand, in Figure 3b, DNase and No DNase. The expression should be unified. To better understand the results, the order of the histograms of DNase+ and DNase- reactions should be the same in Figures 3a and 3b.
3.2.1 NSCLC primary tissues: 3.2.1 NSCLC primary tissues:
First, I wonder if all RNA samples contain the same amount of gDNA. Is the same amount of gDNA contained in the RNA sample?
The PCR with primers, for example with 5’-untranscribed sequence (5’-upstream of TSS)-containing sense one and MALAT1 inner sequence-containing anti-sense one would discriminate the gDNA-templated reaction from cDNA-templated reaction. The primer set only amplifies DNAs with gDNA but not with cDNA. Therefore, this experiment can directly show the gDNA-templated PCR products. I strongly encourage authors to carry out one or two experiments comparing the DNase+ experimental result from the same sample.
If the MALAT1 expression was evaluated as high without DNase treatment but not when it was treated, it suggests that MALAT1 sequence-containing DNAs have been rich. Especially in cancerous cells, it has been shown that extrachromosomal circular DNAs (eccDNAs) are often accumulated. I recommend authors further discuss the reason why false positive was observed.
Reviewer 2 Report
In RT-PCR-based analysis, the authors assumed gDNA contamination is due to the cause of false positives and evaluated the importance of PCR protocol including a DNAse treatment step for the detection of lncRNA samples. In this study, they used MALAT1 as lncRNA and a comparison between “DNAse treatment” and “without DNAase treatment” was performed for a sample of primary tissue and plasma, found that the number of expression/regulation-positives sample for MALAT1 relative to control samples were influenced by the presence or absence of DNAse treatment, and concluded that DNAse treatment was necessary. This theme shows the possibility of false positives due to contamination of gDNA, which is one of the crucial issues for RT-PCR method, and its appropriate improvements in the PCR procedures are necessary. In addition, this theme is recently drawing attention because the PCR method is widely used for detection of COVID-15, so that the study for solving the issues for the method is attractive for the medical research fields not but also for social needs. I believe this kind of study is worthwhile for publications. However, regarding the results in Table 3 and its explanations, it is not still convincing. Since the number of samples was small, it was difficult to determine if their claim on the result is valid. In order to help understanding the results, it seems necessary that further data and its explanation on the PCR primary results such as Cq, ΔCq, ΔΔCq, and so on. Therefore, it is necessary to revise their manuscripts in the following points.
- The abbreviation for HD is not written. It should be specified as Healthy donors (HD).
- The figures and characters in Figure 1 are not clear. Please revise it.
- The purpose and explanations of the results for measuring each sample (tissues and plasma) should be stated in the text. They demonstrated the results for each sample, but explanation on the purpurs and the explanation on the differences in the MALAT1 expression between the two samples are missing.
- About Table 3:
(4-1) It is difficult to understand the Table 3 values and its meaning. For example, in a result of PCR of Primary Tissues, the value “3” of the “Overexpression” of the column for Without DNA treatment and the “Overexpression” of the row of for “After DNAase treatment” is described, but I am wondering if the numbers obtained by performing “Without DNA treatment” and “After DNA treatment” are inconsistent. Please revise the table form and added detail explanations such as definitions for each of values in manuscript and captions for Table 3.
(4-2) It appears that the number of samples is small for their claim on the necessity for the DNAse treatment. Please add the description on the statistical superiority of the number of samples such as a t-test in order to support their claims.
(4-3) Regarding the(4-2), the experimental data and its explanation on the PCR primary results, such as Cq, ΔCq, ΔΔCq, and so on, would be helpful for reader’s understanding of their study. The values for “Overexpression” and “Underexpression'' are just the number values for the author's judgment based on their PCR experiments. Actually, the definitions are cited in the previous reports, but the PCR primary results are missing.
Author Response
Reviewer 2
In RT-PCR-based analysis, the authors assumed gDNA contamination is due to the cause of false positives and evaluated the importance of PCR protocol including a DNAse treatment step for the detection of lncRNA samples. In this study, they used MALAT1 as lncRNA and a comparison between “DNAse treatment” and “without DNAase treatment” was performed for a sample of primary tissue and plasma, found that the number of expression/regulation-positives sample for MALAT1 relative to control samples were influenced by the presence or absence of DNAse treatment, and concluded that DNAse treatment was necessary. This theme shows the possibility of false positives due to contamination of gDNA, which is one of the crucial issues for RT-PCR method, and its appropriate improvements in the PCR procedures are necessary. In addition, this theme is recently drawing attention because the PCR method is widely used for detection of COVID-15, so that the study for solving the issues for the method is attractive for the medical research fields not but also for social needs. I believe this kind of study is worthwhile for publications. However, regarding the results in Table 3 and its explanations, it is not still convincing. Since the number of samples was small, it was difficult to determine if their claim on the result is valid. In order to help understanding the results, it seems necessary that further data and its explanation on the PCR primary results such as Cq, ΔCq, ΔΔCq, and so on. Therefore, it is necessary to revise their manuscripts in the following points.
The abbreviation for HD is not written. It should be specified as Healthy donors (HD).
We apologize for this mistake. In the revised manuscript we have added the full term on page 11.
The figures and characters in Figure 1 are not clear. Please revise it.
In the revised manuscript we have improved Figure 1.
The purpose and explanations of the results for measuring each sample (tissues and plasma) should be stated in the text. They demonstrated the results for each sample, but explanation on the purpose and the explanation on the differences in the MALAT1 expression between the two samples are missing.
About Table 3:
(4-1) It is difficult to understand the Table 3 values and its meaning. For example, in a result of PCR of Primary Tissues, the value “3” of the “Overexpression” of the column for Without DNA treatment and the “Overexpression” of the row of for “After DNAase treatment” is described, but I am wondering if the numbers obtained by performing “Without DNA treatment” and “After DNA treatment” are inconsistent. Please revise the table form and added detail explanations such as definitions for each of values in manuscript and captions for Table 3.
We would like to thank the reviewer for this so important comment. In this contingency table we describe the variation of MALAT1 expression after DNase treatment. In the revised manuscript we have added a Supp.Figure 1 in order to make clear the effect of DNAse treatment in the estimation of MALAT-1 expression both in plasma and tissues samples. Moreover, we have changed the captions of Table 3.
(4-2) It appears that the number of samples is small for their claim on the necessity for the DNAse treatment. Please add the description on the statistical superiority of the number of samples such as a t-test in order to support their claims.
As far as we know this is the first study that evaluated the impact of DNAse treatment in the lncRNA expression and especially in MALAT1 which is a well studied lncRNA. We absolutely agree that our findings need to be further evaluated and validated in more samples and using more lncRNAs and we have mentioned this point in the revised manuscript. Moreover, in the revised manuscript on table 3 we have added the results of paired t-test that was performed between samples before and after DNase treatment in MALAT-1 expression.
(4-3) Regarding the (4-2), the experimental data and its explanation on the PCR primary results, such as Cq, ΔCq, ΔΔCq, and so on, would be helpful for reader’s understanding of their study. The values for “Overexpression” and “Underexpression'' are just the number values for the author's judgment based on their PCR experiments. Actually, the definitions are cited in the previous reports, but the PCR primary results are missing.
MALAT1 was expressed in the healthy donors samples. Cq, ΔCq, ΔΔCq have no meaning without the comparison with normal samples. These is the reason why we think that fold change via overxexpression or underexpression values give the most important information.

Round 2
Reviewer 2 Report
Thank you for your sincere rely and the revision on this study.
I am fully satisfied with this manuscript so i willingly recommend the publication for this journal.